# Acupotomy in Korean Medicine Doctors: A Preliminary Survey on Experiences, Perceptions, and Clinical Usage Status

**DOI:** 10.3390/healthcare11182577

**Published:** 2023-09-18

**Authors:** Hyungsun Jun, Sang-Hoon Yoon, Myungseok Ryu, Hyocheong Chae, Hongmin Chu, Jungtae Leem, Tae-Hun Kim

**Affiliations:** 1Department of Diagnostics, College of Korean Medicine, Wonkwang University, Iksan 54538, Republic of Korea; hs14231423@wku.ac.kr; 2Chung-Yeon Korean Medicine Clinic, Seoul 06224, Republic of Korea; chin9yaaaa@khu.ac.kr; 3Korean Medical Society of Acupotomology, Seoul 07206, Republic of Korea; yeonbu16@gmail.com (M.R.); hc.chae@dongguk.edu (H.C.); chhn2443@wku.ac.kr (H.C.); 4Research Center of Traditional Korean Medicine, College of Korean Medicine, Wonkwang University, Iksan 54538, Republic of Korea; 5Hanbang Cardio-Renal Syndrome Research Center, School of Korean Medicine, Wonkwang University, Iksan 54538, Republic of Korea; 6Korean Medicine Clinical Trial Center, Korean Medicine Hospital, Kyung Hee University, Seoul 02447, Republic of Korea

**Keywords:** Korean medicine, acupotomy, online survey, safety, adverse events

## Abstract

Acupotomy is a widely used medical intervention in traditional East Asian medicine, and efforts are being made to improve its effectiveness and safety. As a first step toward establishing more standardized procedural guidelines, a survey was conducted to explore the current clinical practice status and perceived adverse events (AEs) by Korean Medicine (KM) practitioners. The survey was developed via expert consensus and included information on clinical usage, perception, and the AEs experienced. The largest acupotomy society in Korea, which consists of 185 KM doctors, participated in an online survey conducted in September 2021. Of the 185 KM doctors, 107 (57.8%) responded. Musculoskeletal and connective tissue diseases accounted for 80.8% of suggested indications by KM doctors. Regarding the detailed procedure, there were considerable discrepancies between KM doctors. The most frequent acupotomy-related AEs observed by KM practitioners were bruises (77.3%), fatigue (57.7%), pain (51.8%), and hematoma (51.8%). Only 1.8% of the respondents answered that they had experienced severe AEs. Survey respondents answered that the use of imaging devices during acupotomy and the development of clinical practice guidelines are the most necessary policy requirements for promoting the use and ensuring the safety of acupotomy. To the best of our knowledge, this study marks the initial exploration into the KM physicians’ clinical usage status, AEs experienced, and their requests for standardized guidelines and expanded health insurance coverage concerning acupotomy. Further research should include qualitative studies to assess patient experience and prospective observational studies to examine the effects of operator skills and treatment modalities on AEs and adherence.

## 1. Introduction

Acupotomy, a popular therapeutic tool in traditional East Asian medicine (TEAM), was first created by Zhu Hanzhang in 1976. It combines two types of needles: Bongchim (sharp-edged needle) and Pichim (sword-like needle) [1,2,3]. Unlike traditional acupuncture needles, acupotomy features a surgical blade with a flat tip. This modern technique is regarded for its ease of use, effectiveness, and minimal invasiveness. The primary objective of acupotomy is to incise and dissect the soft tissues, including adherent muscles, ligaments, nerves, and blood vessels. Doing so aims to restore the local tissue’s blood circulation and bring it back to its original dynamic state [2,3]. Acupotomy has been used for musculoskeletal conditions such as osteoarthritis, trigger finger, and lumbar disc herniation [4,5,6,7,8]. Moreover, case reports demonstrate the application of acupotomy for diseases other than musculoskeletal conditions, such as irritable bowel syndrome, and the treatment of atrophic scars caused by skin adhesions [9,10].

However, acupotomy is relatively risky compared to other classical acupuncture procedures. In particular, it involves deep penetration into the body’s structures, stimulation of the muscles and periosteum, and, depending on the disease, potentially reaching into the joint cavities and bursae (Appendix A). Consequently, several studies have addressed the safety of acupotomy treatments. These studies have encompassed preliminary investigations into pre-procedural safety checks, procedural protocols, and infection prevention measures within the context of the acupotomy procedure, all aimed at fostering a secure environment and refining the acupotomy technique [11,12,13]. Furthermore, safety studies using medical imaging equipment such as ultrasound and magnetic resonance imaging have been actively conducted to determine safe needling depths for potentially risky acupuncture points [14,15]. A literature review identified real-world clinical side effects, offering accurate practitioner information [2]. A Delphi study set criteria for acupotomy treatment side effects [16]. Additionally, a prospective observational study collected data on side effect types, severity, and incidence rates in clinical contexts while exploring influential clinical factors [17]. However, efforts to minimize acupotomy procedure side effects go beyond individual clinicians. Apart from guideline development, compensating medical professionals for ensuring procedure safety within the health insurance system is crucial. Previous studies on medical costs for acupotomy lacked evaluation of physician workload. While studies have explored KM doctors’ workload, the distinction between general acupuncture and acupotomy is still not clear [11,18].

Developing comprehensive and standardized procedure guidelines is crucial to lower the associated risks sufficiently. Developing guidelines requires understanding clinical practice status, surveying adverse events (AEs) experience, assessing medical staff workload for safe procedures, and grasping policy requirements for compensation within the medical insurance system as perceived by practitioners. These elements ensure acupotomy procedure safety. As a common quantitative research method, surveys utilize structured tools. This approach enhances domain expertise through data collection and analysis, facilitating guidelines development and appropriate compensation determination within the practitioner insurance systems. Despite potential sampling bias, online surveys are cost-effective primary research methods in South Korea due to widespread Internet usage. They enable the anonymous collection of valid responses and opinions free from location or time constraints [19,20].

This online survey study focuses on physicians’ usage of acupotomy in clinical practice, aiming to explore the current utilization patterns, perceptions, and potential enhancements related to acupotomy procedure safety checklists, suitable insurance reimbursement, experiences with AEs, and requirements for safety procedures.

## 2. Methodology

### 2.1. Institutional Review Board Approval and Participant Consent

Before conducting the online survey, this study received approval from the Kyung Hee University Institutional Review Board (KHSIRB-21-RA-374). Before commencing, participants were provided with information regarding the research purpose, methodology, anticipated duration, and rights of the research participants. Participants voluntarily agreed to participate by providing an electronic signature before completing the survey.

### 2.2. Development of the Questionnaire

A questionnaire was developed for acupotomy based on existing research and references [2,11,12,13,18,21,22]. To ensure validity, two Korean medicine doctors (KMDs) with a minimum of five years of clinical experience and two acupotomy specialists with a minimum of three years of acupotomy treatment experience participated. The research objectives shaped the structure of the survey questionnaire targeting the following dimensions: (1) Clinical usage status, (2) Perceptions, and (3) Range of experienced AEs. The questionnaire encompassed detailed categories, including (1) a holistic overview of the procedure process, (2) the clinical applicability of the “pre- and post-acupotomy procedure safety checklist” guideline, perceptions of physician workload, and suitable fees. Additionally, (3) specific categories addressed experienced AEs following the procedure, encompassing local and systemic AEs, including severe instances. A preliminary survey was conducted with three clinical KMDs who did not participate in the questionnaire development to enhance face validity. The feedback from these three clinical KMDs was used to refine the questionnaire, incorporating their suggestions, and making necessary adjustments to ensure the clarity of survey items. The modifications included using terminology commonly employed in clinical settings (Appendix A). Before the survey, a pilot test was conducted on three KMDs who had undergone acupotomy in clinical practice using a mobile environment (Figure 1).

### 2.3. Items of the Online Survey

The questionnaire comprised 42 items divided into five parts, including 31 multiple-choice items and 11 open-ended items. The design allowed for multiple responses depending on the question. Items that allowed for numerical responses were designed to be answered using numeric values.

Each part was structured as follows:(1)Clinical usage status of acupotomy treatment (Q1–15);(2)Perception of the clinical applicability of the “pre- and post-acupotomy procedure safety checklist” guidelines (Q16–20);(3)Perceptions about the workload and appropriate price for the acupotomy (Q21–25);(4)Observed acupotomy-related adverse events and serious adverse events (Q26–33);(5)Demographic information and other collected data (Q34–42);

The detailed original survey questionnaire is provided in Appendix A.

### 2.4. Participants and Survey Administration

The survey was conducted as a web-based questionnaire using the online survey platform SurveyMonkey. This study was conducted in collaboration with the Korean Medical Society of Acupotomology (http://www.acupotomy.kr/, accessed on 16 September 2023), a leading society in the field of acupotomy that utilizes anatomical and physiological knowledge to analyze the structural and functional characteristics of the human body. The Korean Medical Society of Acupotomology has 185 members and is the largest society in Korea regarding acupotomy therapy. All community members were invited to participate by sharing the survey link through their online messenger platforms to minimize potential sampling bias. The survey was conducted from 1 to 14 September 2021. Participants were offered a mobile coffee coupon worth KRW 10,000 as an incentive for survey completion. 

### 2.5. Statistical Analysis Methods

SPSS Windows software version 22 was utilized for statistical analysis, and R version 4.2.3 was used for creating plots. All variables were presented as numbers (percentages) and means (standard deviations). The following analytical methods were used: Nominal data were analyzed using number (n) and rate (%), and continuous data were presented as mean ± standard deviation. When conducting subgroup analysis, continuous data were compared using an independent samples *t*-test (or Wilcoxon rank-sum test if the data were not normally distributed), and the comparison of nominal data was validated using cross-tabulation (chi-square test or Fisher’s exact test), with a significance level of 5%.

## 3. Results

### 3.1. Demographic Analysis

Of the 185 surveyed participants, 111 responded after providing consent. However, some participants did not respond to a few items in the questionnaire, resulting in varying response rates (Appendix A). A total of 107 KMDs completed the survey by responding to all items, resulting in a response rate of 57.8% (n = 107/185) for the study (Appendix A). Of the respondents, 75.2% (82/109) were male, and the average age of the respondents was 38.3 ± 9.9 years old. Regarding clinical experience, the majority (42.2%, 46/109) had less than 5 years of experience, and the average experience was 11.1 ± 9.39 years (Table 1).

### 3.2. Clinical Usage Status of Acupotomy

A question regarding the use of acupotomy on body sites was asked, which allowed participants to select three overlapping sites. Among the respondents, the most chosen sites were the lumbar region (98.2%), cervical region (89.9%), and upper extremities (53.2%). Of the 26 disease codes provided, 21 were related to Musculoskeletal system and connective tissue disorders, M00–M99 (Table 2). The average career in acupotomy practice was 4.06 ± 3.05 years, with a maximum of 29 years. The most selected blade widths (diameters) for acupotomy needle tips were 0.4 mm, 0.5 mm. The depth of insertion, with the majority reaching the muscle layer (54.5%) and periosteum (30.1%), and shallow insertion limited to the dermis (6.4%) had the lowest proportion. The number of insertion points per session ranged from a minimum of 2 to a maximum of 25, with a median of 8 points. The recommended inter-session interval for patients was a median of 5 days (Table 3, Figure 2). Regarding the medical supplies used for infection prevention, utilizing boric acid, surgical caps, antibiotics, and surgical gowns was low. Ice packs (25.5%) were the most frequently selected treatment for pain relief. Simultaneously, 19.1% of respondents reported using local anesthesia cream for surface anesthesia. However, most (58.2%) participants responded that they had not performed any analgesic treatment (Appendix A).

### 3.3. Differences in Acupotomy Practice between Beginners and Experienced Practitioners

In the subgroup analysis based on proficiency, practitioners with less than three years of experience were categorized as beginners (58/110, 52.7%). In contrast, those with four or more years of experience were classified as experts (52/110, 47.3%). A comparison of acupotomy techniques based on proficiency revealed that the beginners had a higher average number of up and down incision actions (3.79 ± 2.48) compared to the experts (2.92 ± 1.53) (*p* = 0.032). Beginners predominantly reached the muscle layer, whereas experts reached the periosteum (Appendix A).

### 3.4. Perceptions of the Safety Checklists and Health Insurance Coverage of Acupotomy

Concerning the previously developed “pre- and post-acupotomy procedure safety checklist,” respondents generally indicated no unnecessary items. Regarding the post-procedure safety checklist, items were commonly confirmed in the clinical setting in the following order: blood vessel damage, patient education, pain, nerve damage, and vital signs (Appendix A). In terms of physician workload, the average time required for acupotomy procedures was 23.64 ± 18.68 min (Appendix A). When compared to acupuncture treatment, respondents reported that the technical difficulty and physical burden of acupotomy were rated as high or very high by 93.7% (103/110) and 72.7% (80/110), respectively. Consequently, 98.2% of respondents stated that an increase in the treatment fee is warranted (Appendix A). When comparing the perceptions of treatment fees based on proficiency, the expert group set higher costs for acupotomy procedures (Appendix A).

### 3.5. Adverse Events and Serious Adverse Events

Among the local AEs reported after acupotomy, more than half of the participants experienced bruising (77.3%), pain (51.8%), or hematoma (51.8%). Other reported AEs included bleeding (40%), sensory and motor nerve damage-related neurological symptoms (15.5%), local infections (3.6%), and cerebrospinal fluid leakage (0.9%). No respondents reported blistering, organ damage, or pneumothorax as AEs. The most common systemic AEs attributed to acupotomy were fatigue (57.7%), followed by autonomic nervous system dysfunctions (30.6%), and respondents who did not experience any AEs (30.6%). None of the participants reported psychiatric disorders or systemic infections. Regarding serious AEs, 98.2% (108/110) of the participants responded that they had not experienced any serious AEs, whereas 1.8% (2/110) responded positively. The reported serious AEs include central and peripheral nervous system damage and local infection. Factors contributing to the occurrence of AEs included insufficient identification of anatomically safe procedure zones (23.4%), incorrect needling depth (17.1%), improper insertion point selection (13.5%), excessive number of up and down incision actions (12.6%), and blade width (12.6%) (Table 4). In the subgroup analysis comparing AEs based on imaging devices, the group that used imaging devices had a significantly lower number of local AEs (Appendix A).

### 3.6. Requirements of Clinicians for Safe Procedures

Opinions were expressed regarding the need to revise the fee paid by patients to reflect the workload of physicians and medical supplies consumed during the acupotomy procedure. Requests for the inclusion of acupotomy in new health technologies were also mentioned. Additionally, medical imaging devices during the procedure enable the identification of anatomically safe zones, reduces AEs, and enhances therapeutic effects through proactive treatment. The use of medical supplies such as local or topical anesthetics (e.g., creams or lidocaine, etc.) and non-steroidal anti-inflammatory drugs for effective procedures, along with the necessity for standardized procedure guidelines to ensure safety, was described. Furthermore, explaining the differences in AEs based on patient body type and specifying safe depths for different body areas is necessary (Appendix A).

## 4. Discussion

### 4.1. Summary of Findings

Acupotomy is commonly used to treat musculoskeletal and connective tissue disorders. The acupotomy procedure varied among respondents regarding specific aspects, and pain relief treatments were generally performed less. AEs related to acupotomy include local and systemic reactions such as bruising, fatigue, pain, hematoma, bleeding, and autonomic nervous system dysfunctions. Only 1.8% of the respondents experienced serious AEs. There is a high demand for imaging devices, pharmaceuticals, and medical supplies to ensure safe procedures and appropriate insurance reimbursement in acupotomy treatment.

### 4.2. Debate

The survey response rate was 57.8%, higher than the overall survey response rate of 2.23–5% among all KMDs conducted previously [23,24,25,26]. The exact number of KMDs utilizing acupotomy and their demographic information have not been accurately collected. Acupotomy, unlike acupuncture, has not yet become widely generalized in Korea, and it remains a relatively new technology that is actively researched [17,27,28,29]. Consequently, a limited number of practitioners still utilize acupotomy. Therefore, we aimed to extract the most suitable sample from the target population of an academic society actively engaged in acupotomy practice and research. Thus, the conclusions should be interpreted based on the perspective of the acupotomy practitioner subgroup rather than generalized to the entire KMDs population. Additionally, considering the potential diversity in viewpoints among acupotomy practitioners within the broader Korean medicine practitioner community, it is important to acknowledge this limitation. Therefore, after the spread of acupotomy, studies are needed to target a more significant number of KMDs.

A systematic review of acupotomy in Korea reported that povidone-iodine was the most used medication for infection prevention. In contrast, surgical caps, surgical gowns, sterilization wraps, boric acid, and antibiotics were used less frequently [2]. This result is consistent with the findings of this study. Opinions were expressed in the survey regarding personal perspectives on the inadequate alignment between reimbursement fees for acupotomy and the costs of necessary medical supplies for infection prevention. Therefore, further policy research on acupotomy procedural fees and material costs is needed.

Acupotomy has a wide procedural scope and involves the insertion of needles ranging from a minimum diameter of 0.35 mm to 1.2 mm, which can cause significant structural damage [21]. In this survey, many respondents reported local AEs such as pain, bruising, hematomas, and bleeding due to vascular damage. However, the subgroup analysis indicated that these side effects can be minimized when utilizing medical imaging devices actively during procedures, highlighting the importance of their active utilization [30]. Regarding systemic AEs, fatigue and autonomic nervous system dysfunctions (headache, nausea, dizziness, sweating, etc.) were most frequently reported. Fatigue and autonomic nervous system dysfunction symptoms have been frequently reported even after regular acupuncture treatments. Bäumler et al. considered fatigue a symptom of autonomic nervous system dysfunction, among other AEs, after acupuncture treatment [31]. In Kim’s report, acupotomy induces a higher stimulation level than regular acupuncture, which increases the likelihood of experiencing such symptoms. These symptoms occurred more frequently during the initial treatment sessions. Therefore, on the first day of treatment, a gentle approach should be adopted, and for mentally tense patients, the number of insertion points should be reduced [32,33]. Furthermore, it is essential to ensure that the patient receives the treatment in the most comfortable position, emphasizing the occurrence of AEs during a seated posture as reported [22,33,34]. Only a small minority of respondents, two individuals (1.8%), experienced serious AEs. However, all patients reported being in the recovery process or fully recovering without lingering effects.

Similar findings were reported when comparing our study results with the inves-tigation of AEs of acupotomy in patients conducted by Jung et al., including bruising, drowsiness, itching, pain, and fatigue. Furthermore, no irreversible damage or life-threatening events were observed after acupotomy therapy [21]. Additionally, upon reviewing existing studies, it was confirmed that AEs to acupuncture and acupotomy were reported similarly [21,33,34,35,36].

### 4.3. Demands and Contributions for Developing Procedural Guidelines and Appropriate Insurance Coverage 

The need for standardized procedural guidelines that can be applied in clinical settings has been identified as a requirement to ensure safety. They should also consider varying degrees of AEs based on body type and provide safe depths specific to different treatment areas. Notably, regarding the factors contributing to AEs, needling depth was selected as the second most frequent choice. The needling depth may vary depending on gender, body mass index, and treatment area. The absence of navigational imaging devices may cause serious AEs [14,15,30]. Therefore, it is necessary to provide guidelines for safety procedures, including potential AEs based on individual patient characteristics, and procedural guidelines utilizing medical imaging devices, including safe needling depths [37]. Furthermore, there is a need to propose an optimal number of insertion points and up–down incision actions that are the safest and least likely to cause AEs. Incorporating the types of side effects, essentials for safe procedures, improvements to the existing checklist, resources required for safe procedures, and the time and effort of medical professionals, we aim to develop guidelines for safe procedures. To this end, we will establish a committee encompassing stakeholders such as practitioners, researchers, patients, educators, and policymakers. We will continue to develop and refine these guidelines through literature reviews, Delphi studies, and clinical research. Additionally, the survey results confirmed that safe acupotomy procedures require more time than conventional manual acupuncture due to technical difficulty. Therefore, to ensure safe procedures, it is essential to establish appropriate compensation and fees within the health insurance system. 

### 4.4. Strengths, Limitations, and Considerations for Further Study

Regarding the noteworthy limitations of this study, first, the survey was conducted among KMDs specializing in acupotomy, representing a subset of practitioners who use this technique. Therefore, it may not fully represent all KMDs’ widespread practice and AEs associated with acupotomy. However, acupotomy treatment is still in its introductory phase in Korea, with the number of acupotomy practitioners being relatively small compared to the overall practitioner population. To maximize representativeness, we conducted a survey targeting Korea’s most prominent and active acupotomy-related academic society. This approach was taken to overcome limitations associated with the generalizability of our research findings. Second, the questionnaire used in this study was developed based on various references and the literature; however, it has not been formally validated. Therefore, it is necessary to create a validated questionnaire with established reliability and validity for future implementation among the entire population of KMDs. While a Delphi study is being conducted to reach a consensus on the definition of AEs related to acupotomy, a more validated questionnaire is required for assessing clinical usage status and perceptions [16]. Third, regarding the utilization of medical imaging devices during the acupotomy procedures, respondents were asked only about the usage of imaging devices if they were available at their workplace. This may have led to an underestimation of the actual rate of reference imaging during procedures, as cases in which external medical institutions were consulted for imaging were excluded. Furthermore, in the subgroup analysis, rather than comparing the frequency of AE occurrences, we compared the number of AE types related to the usage of medical imaging devices, and the distribution between the two groups was not equal. Therefore, there are limitations in the accuracy of the subgroup comparisons. Fourth, it is essential to note the potential presence of recall bias as the respondents relied on their memories to recall past patient complaints. Fatigue and pain, though significant AEs based on patient-reported subjective experiences, might not be mentioned by patients. Survey research involving medical professionals inherently has fundamental limitations when collecting comprehensive data on AEs. Moreover, the primary objective of our survey was not to investigate AEs resulting from acupotomy procedures. Therefore, the researchers of this study conducted another prospective observational study to collect AEs from the patients’ perspective regarding acupotomy and a systematic literature review in Korea [2,17]. To build a solid foundation for understanding acupotomy AEs, well-designed epidemiological studies with real-world data are necessary for the future. Despite these limitations, this study is significant, as it is the first to collect data on acupotomy procedures, experienced AEs, and requirements from practitioners actively utilizing acupotomy in clinical practice. Moreover, the study used an online survey method to enhance the response rate and collected quantitative and qualitative requirements, providing a basis for future policy recommendations.

## 5. Conclusions

This study investigated the status of acupotomy procedures, AEs, and unmet needs of clinicians. No irreversible damage or life-threatening side effects were observed in the participants after acupotomy therapy, and those who used medical imaging devices reported only a few AEs. Clinicians have emphasized using imaging devices to reduce AEs and enable proactive interventions. Furthermore, recognizing the need for developing standardized guidelines and insurance coverage to ensure safe procedures, this study offers real-world data on acupotomy procedures and their clinical usage status among KMDs. These findings serve as fundamental data for the development of standardized procedural guidelines. However, since the validity of the questionnaire has not been adequately verified, and the study focused on a single professional society, caution is needed when interpreting the results of this study. Furthermore, prospective observational studies are needed to evaluate the impact of factors, such as practitioner expertise, the utilization of imaging devices, and treatment procedures, on AEs and compliance.

## Figures and Tables

**Figure 1 healthcare-11-02577-f001:**
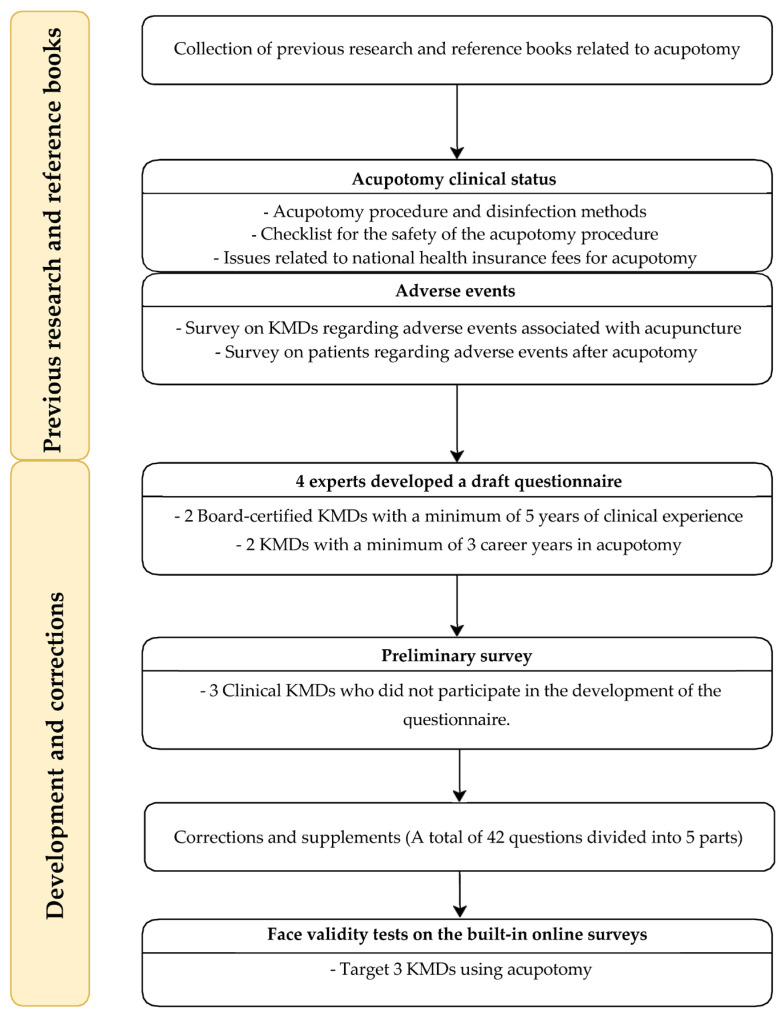
Questionnaire development process. KMDs—Korean medicine doctors.

**Figure 2 healthcare-11-02577-f002:**
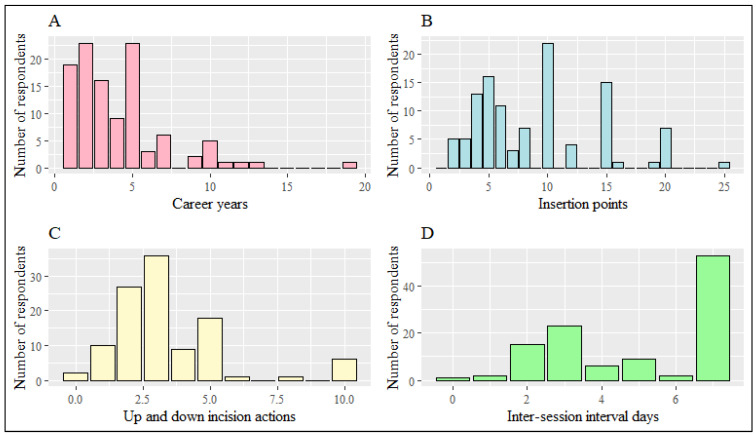
A histogram depicting the number of respondents based on clinical usage status. (**A**), (Q1) career of acupotomy practice (years); (**B**), (Q6) number of insertion points per session; (**C**), (Q7) number of up and down incision actions per point; (**D**), (Q8) recommended inter-session interval for patients (days).

**Table 1 healthcare-11-02577-t001:** Basic characteristics of the survey respondents.

Characteristics (Answered = 109)		n (%)
Gender	Men	82 (75.2)
	Women	27 (24.8)
Age (years)	20–29 s	22 (20.2)
	30–39 s	47 (43.1)
	40–49 s	21 (19.3)
	50–59 s	16 (14.7)
	60–69 s	3 (2.7)
Clinical experience (years)	≤5	46 (42.2)
	6–10	22 (20.2)
	11–20	25 (22.9)
	21–30	8 (7.3)
	31≤	8 (7.3)
Healthcare facility level	General clinic or KM clinic	80 (73.4)
	General or KM hospital	19 (17.4)
	Public health center	10 (9.2)
Board-certified KM doctors	General practitioner	90 (82.6)
	Current training (internship or residency)	8 (7.3)
	Specialist for acupuncture and moxibustion medicine	4 (3.7)
	Specialist for rehabilitation medicine of KM	4 (3.7)
	Specialist for internal medicine of KM	2 (1.8)
	Specialist for ophthalmology, otorhinolaryngology, and dermatology of KM	1 (0.9)
Healthcare facility place *	Urban	97 (89)
	Rural	12 (11)

KM; Korean medicine. * In the administrative divisions of Korea, “Eup” and “Myeon”, located in rural areas, are appropriately translated as “town” and “township”, respectively. Simultaneously, “Dong” in urban regions is suitably rendered as “District” in English contexts.

**Table 2 healthcare-11-02577-t002:** Treatment sites and indications for acupotomy.

Question Number	Treatment Sites		n (%)
Q3	Treatment sites *		109 (100)
		Lumbar and pelvis	107 (98.2)
		Head and neck	98 (89.9)
		Upper extremities	58 (53.2)
		Lower extremities	45 (41.3)
		Back	27 (24.8)
		Abdomen	12 (11)
		Face	10 (9.2)
		Chest	10 (9.2)
Q4	Indications for acupotomy (KCD code) *		107 (100) ^†^
	Head and neck	Headache R51	31 (29)
		Cervicalgia, cervical region M5422	30 (28)
		Cervical disc disorders with cervicalgia M50	29 (27.1)
		Spinal stenosis, cervical region M4802	3 (2.8)
	Face	Bell’s palsy G510	1 (0.9)
	Back	Pain in thoracic spine M546	5 (4.7)
	Lumbar and pelvis	Low back pain, lumbosacral region M5457	37 (34.6)
		Thoracic, thoracolumbar, and lumbosacral disc disorders M51	35 (32.7)
		Spinal stenosis, lumbosacral region M4807	23 (21.5)
		Lumbago with sciatica, lumbosacral region M5447	11 (10.3)
		Degeneration of facet joints M47	9 (8.4)
		Other myalgia, sacroiliac joint M79158	3 (2.8)
	Chest	Thoracic outlet syndrome G540	2 (1.9)
	Abdomen	Indigestion K30	2 (1.9)
	Upper extremities	Other myalgia, shoulder region M79118	19 (17.8)
		Adhesive capsulitis of shoulder M750	9 (8.4)
		Tennis elbow M771	8 (7.5)
		Neuralgia and neuritis, unspecified, forearm M7923	7 (6.5)
		Rotator cuff syndrome M751	5 (4.7)
		Trigger finger M653	4 (3.7)
	Lower extremities	Neuralgia and neuritis, unspecified, lower leg M7926	10 (9.3)
		Other myalgia, lower leg M79168	6 (5.6)
		Gonarthrosis [arthrosis of knee] M17	5 (4.7)
		Other internal derangements of knee M238	2 (1.9)
		Plantar fasciitis M722	2 (1.9)
		Morton’s metatarsalgia G576	1 (0.9)

G, diseases of the nervous system; KCD code, Korean standard classification of diseases code; K, diseases of the digestive system; M, diseases of the musculoskeletal system and connective tissue; R, symptoms, signs, and abnormal clinical and laboratory findings not classified elsewhere. * Duplicate selection was allowed. ^†^ Total number of respondents was 107, and the total number of responses was 299.

**Table 3 healthcare-11-02577-t003:** Features related to acupotomy practice.

Question Number	Features		n (%)	Mean ± SD
Q1	Career of acupotomy practice (years)			4.06 ± 3.05
Q2	Blade width (diameter, mm) *		110 (100)	
		0.35	11 (10)	
		0.4	97 (88.2)	
		0.5	95 (86.3)	
		0.6	50 (45.5)	
		0.75	12 (10.9)	
		0.8	8 (7.3)	
		1	8 (7.3)	
		1.2	8 (7.3)	
Q5	Depth of insertion		110 (100)	
		Muscle	60 (54.5)	
		Periosteum	43 (30.1)	
		Dermis	7 (6.4)	
Q6	Number of insertion points per session			8.95 ± 5.08
Q7	Number of up and down incision actions per point			3.38 ± 2.11
Q8	Recommended inter-session interval for patients (days)			4.98 ± 2.14
Q9	The average number of acupotomy sessions in one week		110 (100)	
		1–5	32 (28.8)	
		6–10	18 (16.2)	
		11–20	21 (18.9)	
		21–30	16 (14.4)	
		31–40	6 (5.4)	
		41–50	9 (8.1)	
		60	3 (2.7)	
		80	2 (1.8)	
		120	1 (0.9)	
		150	1 (0.9)	
		350	1 (0.9)	
		500	1 (0.9)	

SD, standard deviation. * Duplicate selection was allowed.

**Table 4 healthcare-11-02577-t004:** Reported adverse events and serious adverse events after acupotomy.

Question Number	AEs and SAEs		n (%)
Q26	Local AEs *		110 (100)
		Bruise	85 (77.3)
		Pain	57 (51.8)
		Hematoma	57 (51.8)
		Bleeding	44 (40)
		Neurological symptoms	17 (15.5)
		Local infections	4 (3.6)
		None	2 (1.8)
		Cerebrospinal fluid leakage	1 (0.9)
		Blister	0 (0)
		Organ damage	0 (0)
		Pneumothorax	0 (0)
Q27	Systemic AEs *		111 (100)
		Fatigue	65 (57.7)
		Autonomic nervous system dysfunctions (headache, nausea, dizziness, needle sickness, etc.)	34 (30.6)
		None	34 (30.6)
		Psychiatric disorder (anxiety, impatient, etc.)	0 (0)
		Systemic infection	0 (0)
Q28	Main factors for the occurrence of AEs *		111 (100)
		Insufficient identification of anatomically safe procedure zones	26 (23.4)
		Incorrect needling depth	19 (17.1)
		Improper insertion point selection	15 (13.5)
		Excessive number of up and down incision actions	14 (12.6)
		Misconduct in the selection of blade width	14 (12.6)
		Wrong blade direction	11 (9.9)
		Misconduct of needling speed	9 (8.1)
		Negligence in the contraindication for acupotomy	8 (7.2)
		Negligence in the disinfection and patient education (preventing infection, pain control) after acupotomy	6 (5.4)
Q29	Experience of SAEs		110 (100)
		No	108 (98.2)
		Yes	2 (1.8)
Q30	SAEs *		2 (100)
		Central, peripheral nervous system damage	1 (50)
		Local infections	1 (50)
Q31	Main factors for the occurrence of SAEs *		2 (100)
		Insufficient identification of anatomically safe procedure zones	1 (50)
		Improper insertion point selection	1 (50)
		Negligence in the disinfection and patient education (preventing infection, pain control) after acupotomy	1 (50)
Q32	Consequences of SAEs *		2 (100)
		Complete recovery	1 (50)
		Recovering	1 (50)

AEs, adverse events; SAEs, serious adverse events. * Duplicate selection was allowed.

## Data Availability

The data that support the findings of this study are available from the corresponding author upon reasonable request.

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
