# Peer review of "Acupotomy in Korean Medicine Doctors: A Preliminary Survey on Experiences, Perceptions, and Clinical Usage Status"

_healthcare, 2023, doi:10.3390/healthcare11182577_

Round 1

Reviewer 1 Report

This study, a preliminary survey on experience, perceptions, and clinical usage status is interesting. I would like to give some comment for the better paper.

1. Abstract L24. Please correct 'doctor' to KM doctors

2. Introduction L59. What does advancing knowledge mean?

3. The purpose of this study is not clear. Please suggest specific purpose of this study. In L89~94, there are 1 to 5 structures of the questionnaire. It seems that it coincided with the results. This seems to be your specific purpose.

4. Results. Please reorder all your tables. For example, in table 1, please make another column for sub variables. It would be more visual if you suggest in bar chart (this is only recommendation).  

5. Results L220~L 242. In the questionnaire, the items were about finding the perception of KM doctors of safety checklist and the workload. The sentence should be revised into how they perceived the necessity of the safety checklist and to reflect their workload to the fee.

6. Discussion. Please include how it is to develop guideline reflecting the results of this survey.

Reviewer 2 Report

Thank you for inviting me as a reviewer of this valuable manuscript. I recommend following suggestions for improving the quality of manuscript.

(Comment 1) The introduction section is too short. I recommend authors to add a '1.1. Literature Review Section' on Acupotomy as a sub-section

(Comment 2) The author conducted a survey on 185 members of the Korean Medical Society of Acupotomology. I think this number is insufficient to suggest that it represents Korea's acupotomy. This point must be discussed in Discussion Section.

(Comment 3) The authors described the questionnaire development process in detail. However, it is difficult to suggest that the reliability and validity of the questionnaire have been verified. This point must be discussed in Method Section and Discussion Section. Regarding Method Section, I recommend authors to supplement the key opinions and surveyitem changes during the questionnaire development process.

(Comment 4) AEs were answered by an acupotomy practitioners. In particular, fatigue or pain are subjective complaints of patients, so a patient survey is needed. This point is limitation of this study.

(Comment 5) (line 164) Urban, 97 (89); Rural, 12 (11). What is the criteria for classifying Urban and Rural?

(Comment 6) (line 290) 'Policy demands' is not an appropriate term. Demands for standardized procedural guidelines would be more appropriate.

None

Reviewer 3 Report

I reviewed the paper titled ' Acupotomy in Korean Medicine doctors: A preliminary survey on experiences, perceptions, and clinical usage status', in which the authors studied the safety and efficacy of acupotomy based on filling questionnaires from providers with past experience. This sounds interesting, however, there are some remarks that should be considered:

1- As mentioned in the discussion, there are significant limitations including but not limited to recall bias. On the other hand,  there may be many minor or major side effects that patients might forget or miss to mention to their providers and that could be better to also have patients' ideas beside providers' comments. 

2- Results section have been written confusing and is better not to repeat all details that are already mentioned in the tables. 

3- The questionnaire used is a kind of invalidated checklist that results and conclusions should be mentioned cautiously. 

The English of the text is good. 

Round 2

Reviewer 1 Report

The revision manuscript well reflected the reviewer's need. However, Please reorder tables 1,2,3, 4.  please make another column for sub variables. And please change ratio into rate.  frequency(rate) ->n(%)

Author Response

Thank you for your comments. Our research team also deliberated on the tabular format. In our efforts to reduce table size due to the extensive content, we realized that it might have compromised clarity. Following your suggestion, we have added additional column to the table for a more organized presentation, and we have also revised the supplementary table accordingly.

Furthermore, we have updated all instances of 'frequency' to 'number' and changed 'ratio' to 'rate.' In the tables, we have modified it to 'n (%)'.

We appreciate your valuable feedback, and these changes have been made in response to your important suggestions.

Please see the attachment file.  

Reviewer 3 Report

My comments have been addressed. 

Author Response

Thank you.